# Evaluation of Life Quality of Patients Submitted to Cataract Surgery with Implantation of Trifocal Intraocular Lenses

**DOI:** 10.3390/jpm13030451

**Published:** 2023-02-28

**Authors:** Suowang Zhou, Ana Galrão de Almeida Figueiredo, Adilamu Abulimiti, Wilson Takashi Hida, Xu Chen

**Affiliations:** 1Aier School of Ophthalmology, Central South University, Changsha 410083, China; 2HOB-Hospital Oftalmológico de Brasília, Brasília 70200-670, Brazil; 3Department of Ophthalmology, Shanghai Aier Eye Hospital, Shanghai 200336, China; 4Ophthalmology, HOB-Hospital Oftalmológico de Brasília, Brasília 70200-670, Brazil; 5New Bund Medical and Surgical Center, Sino United Health Clinics, Shanghai 200011, China; 6Department of Ophthalmology, Shanghai Aier Qingliang Eye Hospital, Shanghai 201799, China

**Keywords:** trifocal intraocular lens, cataract surgery, subjective questionnaire, visual quality

## Abstract

This study aimed to evaluate the quality of life and the satisfaction level of Brazilian and Chinese patients who underwent cataract surgery for Acysof IQ PanOptix Model TFNT00 (Alcon Laboratories, Fort Worth, TX, USA) implantation. This retrospective study enrolled 51 patients from China and 51 patients from Brazil. At the 3-month follow-up, uncorrected distance visual acuity (UDVA) at 5 m, uncorrected intermediate visual acuity (UIVA) at 60 cm, and uncorrected near visual acuity (UNVA) at 40 cm were evaluated; Catquest 9SF and the Near Activity Visual Questionnaire (NAVQ) were administered to the patients. The results revealed that the Brazilian patients gained better UDVA and UNVA (*p* < 0.001), while the Chinese patients gained better UIVA (*p =* 0.001). With regards to the patients’ overall satisfaction with their current vision, the Brazilian patients scored higher (*p* = 0.002). In situations related to distant and near vision, the Brazilian patients scored higher, while in situations related to intermediate vision, the Chinese patients scored higher. No differences were found between the gender or age subgroups, but the normal axial length (AL) subgroup showed the highest level of satisfaction (*p* = 0.002). The patients implanted with TFNT00 IOL obtained excellent objective and subjective outcomes in both cultures. The Brazilian patients showed higher satisfaction with their distant and near vision, while the Chinese patients were more satisfied with their intermediate vision.

## 1. Introduction

With recent advances in intraocular lens (IOLs) technology, cataract surgery has evolved from a form of rehabilitation surgery to a refractive procedure [1]. The IOL used for the replacement of opaque lenses plays a vital role in achieving the desired visual outcomes after surgery [2]. Multifocal IOLs (MIOLs) are designed to allow unaided good vision across various distances by providing multiple foci simultaneously, allowing greater spectacle independence after cataract surgery [3,4]. These MIOLs can be bifocal or trifocal. The first MIOLs were introduced in the late 1980s, providing spectacle independence by correcting near and distance vision [5]. However, with changes in lifestyle and work, the demand for skills reliant on intermediate vision, such as using computers, laptops, and mobiles, increased greatly. Trifocal IOLs were developed, offering major contributions to the realization of a full range of vision and presbyopia correction, with an additional focal point for intermediate vision.

There are different commercially available multifocal IOLs with different focal points for long-distance, intermediate, and near vision. The Acysof IQ PanOptix Model TFNT00 (Alcon Laboratories, Fort Worth, TX, USA) is a one-piece aspheric hydrophobic trifocal IOL, launched in 2015. It is characterized by its 60-cm intermediate focal point, which is a more natural and comfortable working distance from which to perform functional tasks on computers and mobile phones. Many studies show that this latest trifocal IOL achieved encouraging results for long-distance, intermediate, and near vision [6,7,8]. Recently, clinicians have gradually realized that visual acuity alone is inadequate for vision-quality assessment. Even patients with good visual acuity may complain of discomfort in daily activities [9]. The characteristics of lifestyle and work also affect patient satisfaction, especially across different cultures and racial groups. Vision-related quality of life (QoL) parameters might not correspond to visual acuity because acuity does not always reflect other aspects of vision, such as contrast issues and photopic symptoms, such as halos or glare, as well as visual performance in daily activities.

Quality of life embraces a range of physical and psychological criteria that describe an individual’s functional ability. A number of questionnaires have been developed to assess patients’ vison-related QoL or to survey their health status [10,11]. In the present study, we applied a standardized questionnaire specifically used to determine the QoL and satisfaction of patients who underwent cataract surgery with trifocal IOL implantation between Brazilian and Chinese populations. To the best of our knowledge, this is the first study to investigate different subjective and objective results in different culture populations.

## 2. Materials and Methods

This was a retrospective comparative case series, based on the analysis of 51 Chinese patients and 51 Brazilian patients. The study was conducted in accordance with the tenets of the Declaration of Helsinki and received the approval of the hospital ethics committee (the approval number: SHAIER2021IRB02). We reviewed the patients who underwent cataract surgery and implanted the new trifocal IOL (AcrySof IQ^®^ PanOptix IOL TFNT00, Alcon Laboratories, Inc., Fort Worth, TX, USA) from September 2020 to June 2021. Patients with complete preoperative and postoperative examination were included. Preoperatively, the patients were submitted to a complete ophthalmologic evaluation, including: slit lamp exam, intraocular pressure (IOP), evaluation of the refractive status, ocular biometric measurements, and fundoscopy. Patients with a history of previous ocular surgery or coexisting ocular pathologies, such as glaucoma, macular degeneration, and severe dry eye and patients with corneal astigmatism of >1.0 diopter (D) were excluded from the study. Furthermore, in order to better evaluate the performance of this new IOL, we also excluded patients with any IOL-independent complications during the cataract surgery or the postoperative follow-up. The surgeries were performed by surgeon X. Chen (China) and surgeon Hida W.T. (Brazil) through standard phacoemulsification cataract procedure. The IOL power was calculated using optical biometry (IOL-Master 700; Carl Zeiss Meditec, Jena, Germany) and Barret formulas. The target refraction was plano, or the value closest to plano.

At 3 months after surgery, the following parameters were measured: uncorrected distance visual acuity (UDVA) at 5 m, uncorrected intermediate visual acuity (UIVA) at 60 cm, uncorrected near visual acuity (UNVA) at 40 cm, best corrected visual acuity (BCVA), and manifest refraction spherical equivalence (MRSE), as well as complete slit lamp, intraocular pressure and fundus exams. Visual acuity testing was performed with international standard visual acuity table under photopic light conditions (85 cd/m^2^). The results were converted to logarithm of the minimum angle of resolution (logMAR) to make the results comparable to those of other studies. Furthermore, these patients were administered questionnaires to assess visual quality and satisfaction with life. The applied questionnaires were: Catquest 9SF, which is used to assess the benefits of cataract surgery and was recently validated in a Spanish population [12]; and the Near Activity Visual Questionnaire (NAVQ), to evaluate the level of difficulty experienced in performing various activities of daily life. The questionnaire comprises 20 questions, of which 2 explore overall satisfaction with current vision, 1 reveals the overall level of difficulty caused by vision in daily life, and the remaining 17 are various tasks that are common in daily life. The Chinese and Portuguese versions of the questionnaire were administered to the Chinese and Brazilian patients, respectively. All data are included on Medoms online platform. There are 5 options for each question: 1 = Yes, very great difficulty, 2 = Yes, great difficulty, 3 = Yes, some difficulty, 4 = No, no difficulty, 5 = Cannot decide (except for question 3 and question 20, for which 1 = very dissatisfied, 2 = generally acceptable, 3 = satisfied, 4 = quite satisfied, 5 = very satisfied). For question 3 and question 5, we calculated the average scores directly. Higher scores meant greater satisfaction, with 5 as the highest; for other questions, we first removed option 5 and then obtained the average scores; therefore, 4 was the highest level. The scores of the questionnaire were calculated and compared between Brazil and China and between different subgroups.

All statistical analyses were conducted with SPSS, version 22 (IBM, New York, NY, USA). Independent-sample t-tests were used to analyze differences in questionnaire scores between Brazil and China. One-way ANOVA was used to analyze differences among subgroups. All tests of association were considered statistically significantly at *p* < 0.05.

## 3. Results

### 3.1. Main Outcomes

#### 3.1.1. Demographic Information and Postoperative Visual Acuity

This study comprised 102 patients (51 from Brazil and 51 from China). Table 1 displays the demographic information and mean preoperative and postoperative clinical data obtained in each group and the statistical significance of the differences between the groups for each parameter evaluated. The mean ages of the Brazilian and Chinese patients were significantly different (*p* = 0.007); 61.29 ± 7.43 years and 57.66 ± 11.22 years, respectively. Furthermore, significant differences were detected between the two groups in terms of AL, ACD, and IOL power (all *p* < 0.001). The Chinese patients showed longer AL (26.28 ± 2.32 mm vs. 23.22 ± 1.45 mm), deeper ACD (3.37 ± 0.38 mm vs. 3.13 ± 0.35), and lower IOL power (14.94 ± 5.42D vs. 21.46 ± 2.49D).

Table 2 summarizes the 3-month postoperative visual acuity and refractive data. The mean values of the overall UDVA, UIVA, and UNVA in this study were 0.03 ± 0.07 logMAR, 0.05 ± 0.09 logMAR, and 0.04 ± 0.08 logMAR, respectively. Furthermore, statistically significant differences were found between Brazil and China in terms of the UDVA, UIVA, and UNVA (*p* ≤ 0.001). The Brazilian patients gained better postoperative UDVA (0.0097 ± 0.033 logMAR vs. 0.0573 ± 0.092 logMAR) and UNVA (0.0175 ± 0.05 logMAR vs. 0.0660 ± 0.09 logMAR), while the Chinese patients gained better postoperative UIVA (0.0297 ± 0.09 logMAR vs. 0.0728 ± 0.09 logMAR). In total, 99% and 88% of the patients achieved a postoperative monocular UDVA of 0.10 logMAR or better in the Brazil and China groups, respectively (*p* = 0.005; Figure A1). In addition, 79% and 92% of the patients achieved a postoperative monocular UIVA of 0.10 logMAR or better in the Brazil and China groups (*p* = 0.006). Furthermore, 94% and 84% of the patients achieved a postoperative monocular UNVA of 0.10 logMAR or better in the Brazil and China groups, respectively (*p* = 0.024). Finally, 100% and 96% of the patients achieved a postoperative monocular BCVA of 0.10 logMAR or better in the Brazil and China groups, respectively (*p* = 0.121).

#### 3.1.2. QoL Questionnaire Results

For overall satisfaction, 63 patients (62%) reported that their vision caused them no difficulties in their daily lives. Furthermore, 27 patients (26%) reported having some difficulties. Regarding their current vision, sixty patients (59%) were very satisfied, thirty-five patients (34%) quite satisfied, and five patients (5%) felt unsatisfied. Of these five unsatisfied patients, four (80%) patients reported great difficulty in reading tiny prints, such as newspaper text, two (40%) patients experienced difficulties in performing handicrafts or seeing nearby objects in dim light. Figure A2 displays the patients’ answers to the questionnaire (Q3–Q19), which explores difficulties in performing different activities in daily life. In total, 90% of the patients reported no difficulty in recognizing people’s faces and engaging in an activity or hobby, such as playing card games, or gardening. Furthermore, 66.5% of the patients reported no difficulties when reading small text, 69.4% of the patients reported no difficulty in maintaining prolonged work using nearby objects, and the remaining patients reported some or great difficulties. In addition, the satisfaction levels relating to the remaining tasks ranged between 70% and 90%. 

Table 3 and Figure 1 show a comparison of the QoL questionnaire scores at 3 months after the implantation of the new trifocal IOL (Acysof IQ PanOptix Model TFNT00) in the Brazilian- and Chinese-patient groups. The Brazilian group was associated with better satisfaction in activities such as reading text in newspapers ( *p* < 0.001), viewing the prices of goods when shopping (*p* = 0.002), and using their sight to walk on uneven surfaces (*p* = 0.043). For the general satisfaction-related QoL items, such as whether, at the time the questionnaire was administered, their sight in some way caused them difficulties in everyday life, whether they were satisfied with their sight, and how satisfied they were with their near vision, the Brazilian group of patients also showed higher levels of satisfaction (*p* = 0.031, *p* = 0.002, *p* = 0.001). The Chinese group tended to gain better outcomes in activities involving the reading of small-print items, such as newspaper articles, the items on a menu, and telephone directories (*p* = 0.031).

#### 3.1.3. Comparison of QoL Questionnaire Scores between Different Subgroups

We calculated the total score for each question and made comparisons between different subgroups, such as gender, age, and AL. No differences were found between the gender (*p* = 0.605; Figure 2) and age groups (*p* = 0.353; Figure 3). Furthermore, the patients were divided into four groups according to AL: group 1 (AL < 22 mm); group 2 (22 mm ≤ AL < 24 mm); group 3 (24 mm ≤ AL < 26 mm); group 4 (AL ≥ 26 mm). Significant differences were found between the AL subgroups (*p* = 0.002), and the patients with normal AL (22 mm ≤ AL < 24 mm) scored the highest (Figure 4).

## 4. Discussion

Cataract surgery with an IOL implantation has the potential to improve a patient’s acuity and refine the refractive error to a given target. Although bifocal IOLs provide good visual function over large and short distances, intermediate vision also plays a vital role in activities of daily life, such as using laptops or focusing on the dashboard while driving. Therefore, trifocal IOLs are used increasingly widely around the world, with validated outcomes in spectacle freedom and patient satisfaction [7,13,14,15]. In order to achieve spectacle independence and ideal patient satisfaction, a good knowledge of patients’ lifestyle characteristics and the demands on their vision is of great importance.

It is recommended that ophthalmologists to choose the correct multifocal IOL type depending on what their patients do or where they live. Furthermore, different cultures involve different visual requirements through lifestyles and work. Among Asian people, significant time may be spent using objects at short distances, such as computers, tablets, and mobile phones for work, or reading books during leisure time. By contrast, Western populations may lead more outdoor lives. In particular, Chinese text may be very small and intricate compared to English characters; hence a full reading aid is usually needed. Furthermore, Asian people are generally shorter in stature, with shorter arms, which creates shorter distances between individuals’ faces and their books, mobile phones, and other materials. Low-add multifocal IOLs or normal monovision strategies may not be able to cope with the demands of reading among Asian people. The PanOptix IOL has a novel diffractive structure and divides the incoming light to create intermediate- and near-add powers of +2.17 diopters (D) and +3.25 D, respectively. This new trifocal IOL is equivalent to bifocal IOLs in photopic short- and long-distance performance while providing an optimal intermediate focus at 60 cm, which is assumed to be more in line with the reading habits of the Chinese population [16]. Recently, the good visual performance of PanOptix IOL was confirmed [7,14,15,17]. Since the real success of a medical intervention should be measured by its effects on patients’ QoL, we specifically evaluated the QoL and satisfaction of patients who underwent cataract surgery and the implantation of PanOptix IOL, using the Catquest 9SF questionnaire and NAVQ to assess their near-vision satisfaction.

In this study, the mean values of the overall UDVA, UIVA, and UNVA were 0.03 ± 0.07 logMAR, 0.05 ± 0.09 logMAR, and 0.04 ± 0.08 logMAR, respectively, which were higher than or similar to those described for other trifocal lenses [18,19,20,21,22,23]. In particular, we found that 99% of the Brazilian patients and 88% of the Chinese patients in this study gained a postoperative UDVA of 0.10 logMAR or better, which was higher than in previous studies. We consider this to be partially related to the differences between the exclusion criteria. In order to better evaluate the performance of the IOL, we excluded patients with any IOL-independent operative or postoperative complications. One of the concerns about trifocal technology is that the light distribution required to create an intermediate focus might interfere with the distant and near focuses and reduce visual acuity. However, the uncorrected distance and near vision achieved by our patients were similar to those in other studies with bifocal lenses [24,25,26], suggesting that the addition of an intermediate focus does not interfere with the other two focuses. Furthermore, a comparison of the visual outcomes obtained with PanOptix IOL was performed. Regarding the distant-, intermediate-, and near-vision outcomes, statistically significant differences were found between the Brazil and China group: the Brazilian population gained better UDVA and UNVA, while the Chinese population gained better UIVA. We consider this difference to be partially correlated with postoperative refractive errors: the postoperative SE of the Chinese patients was significantly higher than that of the Brazilian patients (*p* = 0.018). This factor can directly affect distant vision. Another factor is that the enrolled Chinese patients had stronger myopic characteristics than the Brazilian patients. These highly myopic patients may have experienced poor retinal or macular function, which would have led to slightly poor uncorrected vision, and even best corrected vision.

In addition to VA and refraction, the visual quality outcomes were also evaluated in our study with a validated questionnaire. In our study, reading small newspaper text, seeing objects in dim light, and maintaining a focus on prolonged work with nearby objects were the three most challenging tasks. For each of these categories, 34 (33%), 27 (26%), and 21 (21%) patients reported having some or great difficulties, respectively. Ahmet et al. [27] came to a similar conclusion in a recent study, in which they applied the National Eye Institute Visual Function Questionnaire-14 (VF-14 QOL questionnaire). They found that reading small print was the most challenging task and that binocular implantation was associated with improvements in vision-related QOL when compared with monocular implantation.

In a prospective case series comprising 48 patients, Jorge et al. [17] used the Catquest 9-SF questionnaire to evaluate patients’ vision-related QOL. They pointed out that most of their patients reported little or no difficulty with the activities included in the questionnaire; driving at night was the most challenging task, with 26% of the patients reporting difficulties occasionally or often. Kohnen et al. [7] performed a short quality-of-vision evaluation through the National Eye Institute Visual Function Questionnaire-14 (NEI VFQ-15) and found driving at night to be the most difficult task. Poor night vision is another concern regarding trifocal technology. This might be related to the perception of halos and glare in photopic conditions and energy utilization in mesopic conditions. Fortunately, dysphotopsia gradually diminishes with the visual neuroadaptation process, which is necessary for the human brain to adapt to the different images that are provided by multifocal optics [15]. The typical neuroadaptation process after multifocal IOL implantation involves 3 months to 1 year [28]. Thus, detailed preoperative explanation and instructions on the correct use of illumination helps patients to adapt to the new vision created by IOL.

Furthermore, we compared the overall questionnaire scores between the Brazil and China groups. We noted that the Brazilian patients’ scores were statistically higher for questions reflecting overall satisfaction with short-distance vision, such as“Q2: Are you satisfied or dissatisfied with your sight at present?”, and long-distance vision, including “Q3: Reading text in newspapers” and “Q6: Seeing to walk on uneven surfaces.” By contrast, the Chinese patients’ scores were statistically higher for “Q10: Do you find any difficulties in reading small print, such as newspaper articles, items on a menu, telephone directories.” Interestingly, we observed that the Chinese population now rarely reads print newspapers or tiny telephone directories. Apps such as Wechat, Tiktok, and others are more popular among these Chinese patients, and voice or video calls and friend contacts via Wechat has replaced traditional telephone communication and telephone–address books. Occasionally, elderly individuals may be seen reading newspaper presbyopic glasses. However, our patients, whose average age was 57 years, were more likely to browse news and messages or search for contacts on their phones or tablets with larger fonts. Furthermore, for the items on a menu, a discrepancy was also observed between the respective cultures: Chinese menus usually have larger font sizes than English/Portuguese menus, and feature corresponding pictures. These factors generally make reading menus easier for elderly people in China. Hence, although Q10 was designed to reflect nearby scenes in daily life for Brazilian patients, it actually depicted intermediate-vision scenes when applied in Chinese because of the difference in culture. In general, the Brazilian patients showed higher levels of satisfaction with their near and distant vision, and the Chinese patients showed higher levels of satisfaction with their intermediate vision. These findings were consistent with the VA outcomes. With regards to this VA difference, we consider that in eyes with shorter AL during reading, there may be a greater shift in the IOL toward the cornea. This may account for the better near vision in the Brazilian individuals.

In order to further investigate the satisfaction levels among populations with different demographic characteristics, we also conducted a subgroup analysis based on gender, age, and AL. No significant differences were found in the gender- and age-subgroup analyses, indicating that this new trifocal IOL type suits populations with wide age ranges. The youngest patient enrolled in this study was 25 years old, the oldest was 82 years old, and both patients demonstrated excellent outcomes.

With regards to the AL, the patients were divided to four subgroups. In Brazil, significant differences were found between the four subgroups, and the patients with ALs longer than 28 mm scored the highest. However, due to the relatively low prevalence of significant myopia in Brazil, only one patient was enrolled with an AL longer than 28 mm. We considered that there may be have been individual bias and integrated patients from both countries. In this case, the scores also significantly differed; the normal-AL group (22 mm ≤ AL < 24 mm) scored the highest and the short-AL group (AL < 22 mm) scored the lowest. To further explore the reasons for the poor satisfaction of the short-AL patients, we checked the questionnaire results of this set of patients and found that they showed their greatest levels of dissatisfaction on Q3 (reading text in newspapers) and Q10 (reading small prints), both of which are near-vision tasks. Hence, we compared the UNVAs between the different AL subgroups. In contrast to our prediction, the short-AL group did not display the worst UNVA. Instead, the short-AL group had a better UNVA than the patients with ALs longer than 26 mm. We consider this divergence between objective and subjective outcomes to be related to reading habits. Unlike the short-AL patients, the myopic patients were used to performing working tasks at a relatively short distance. Hence, reading small prints at near distance is a habitual routine for them. Furthermore, previous studies reported that hyperopic eyes have larger Kappa angles and higher levels of corneal higher-order aberrations, which may deteriorate visual quality. Based on this, careful preoperative assessment is recommended for trifocal IOL implantation in hyperopic eyes.

The study has several limitations. One is the differences in the baseline information. Significant differences were present between the ages and ALs of the Chinese and Brazilian patients, which also potentially exerted an impact on the satisfaction assessment. On the other hand, more comprehensive assessments of near vision, such as the reading of speed measurements, were difficult to implement in this study, mainly because it is difficult to compare the reading speed between Chinese and Portuguese due to the different characters in the two languages and the respective cultures’ reading habits. Last, limited by the retrospective nature of the study, we only collected the dysphotopsia information from the Chinese group, which revealed that halos are the most common form of discomfort after the implantation of PanOptix IOL; nearly half of the patients occasionally experienced halos.

## 5. Conclusions

The patients implanted with trifocal PanOptix IOL obtained excellent VA outcomes for long, intermediate, and short distances both in Brazil and in China. The results of the Catquest 9-SF and the QNVA questionnaire confirmed that spectacle independence can be obtained for distant and near activities, as well as activities requiring intermediate vision. Few difficulties were reported for reading items with very small print, such as newspapers. The Brazilian patients showed higher satisfaction levels with their distant and near vision, while the Chinese patients were satisfied with their intermediate vision.

## Figures and Tables

**Figure 1 jpm-13-00451-f001:**
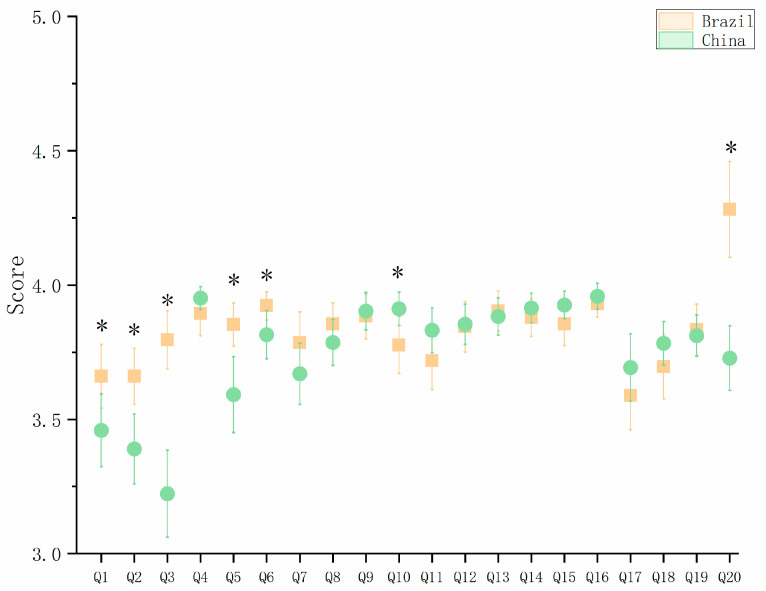
Quality-of-vision scores obtained with the questionnaire (* means significant difference was found between the two groups).

**Figure 2 jpm-13-00451-f002:**
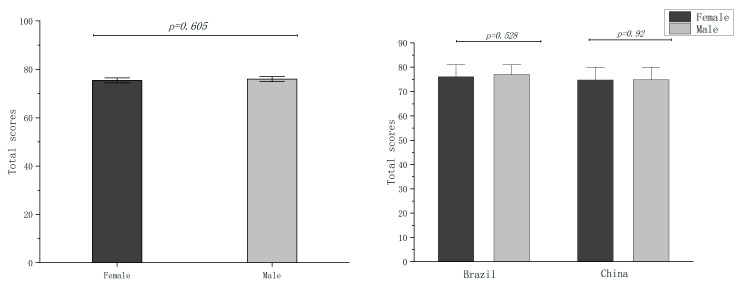
Comparison of total scores between genders.

**Figure 3 jpm-13-00451-f003:**
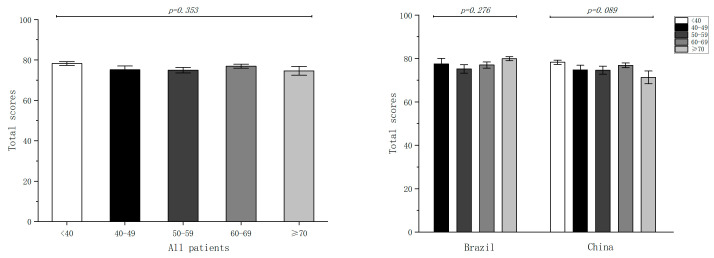
Comparison of total scores between age groups.

**Figure 4 jpm-13-00451-f004:**
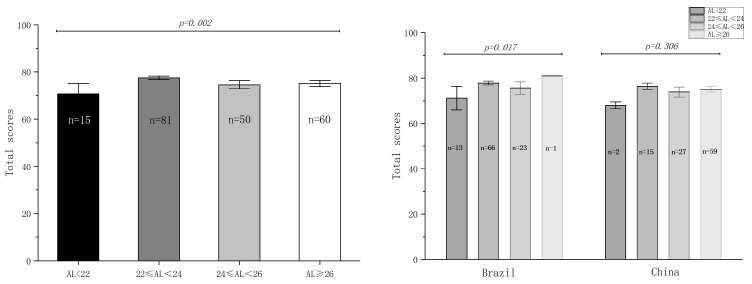
Comparison of total scores between AL groups.

**Table 1 jpm-13-00451-t001:** Demographic information.

	Brazil	China	*p*
Age	61.29 ± 7.43	57.66 ± 11.22	0.007
Gender			
Male	45	46	
Female	58	57	
AL	23.31 ± 1.04	26.28 ± 2.32	0.000
ACD	3.13 ± 0.35	3.37 ± 0.38	0.000
IOL power	21.46 ± 2.49	14.94 ± 5.42	0.000

AL = axial length; ACD = anterior chamber depth; IOL = intraocular lens.

**Table 2 jpm-13-00451-t002:** Mean 3-month postoperative visual acuity and refractive data.

	Overall	Brazil	China	*p* †
UDVA (LogMAR)	0.03 ± 0.07	0.01 ± 0.03	0.06 ± 0.09	*0.000*
UIVA (LogMAR)	0.05 ± 0.09	0.07 ± 0.09	0.03 ± 0.09	*0.001*
UNVA(LogMAR)	0.04 ± 0.08	0.02 ± 0.05	0.06 ± 0.09	*0.000*
BCVA (LogMAR)	0.02 ± 0.04	0.01 ± 0.03	0.02 ± 0.06	*0.021*
SE (D)	−0.14 ± 0.46	−0.06 ± 0.37	−0.21 ± 0.52	*0.018*
DS (D)	0.08 ± 0.47	0.14 ± 0.35	0.02 ± 0.55	*0.079*
DC (D)	−0.43 ± 0.46	−0.40 ± 0.45	−0.47 ± 0.47	*0.273*

LogMAR = logarithm of the minimum angle of resolution; D = diopter; UDVA = uncorrected distance visual acuity; UIVA = uncorrected intermediate visual acuity; UNVA = uncorrected near visual acuity; BCVA = best corrected visual acuity; SE = spherical equivalence; DS = sphere diopter; DC = cylinder diopter; † *p* value of the Student test between Brazil and China groups.

**Table 3 jpm-13-00451-t003:** Quality-of-life questionnaire scores for Brazil and China.

Questionnaire Items	Brazil	China	*p*
Q1: Do you find that your sight at present in some way causes you difficulty in your everyday life?	3.66 ± 0.607	3.46 ± 0.691	0.031
Q2: Are you satisfied or dissatisfied with your sight at present?	3.66 ± 0.534	3.39 ± 0.665	0.002
Q3: Reading text in newspapers?	3.8 ± 0.549	3.22 ± 0.828	0.00
Q4: Recognizing the faces of people you meet?	3.89 ± 0.418	3.95 ± 0.216	0.211
Q5: Seeing the prices of goods when shopping?	3.85 ± 0.408	3.59 ± 0.72	0.002
Q6: Seeing to walk on uneven surfaces, e.g., cobblestones?	3.92 ± 0.269	3.82 ± 0.459	0.043
Q7: Seeing to do handicrafts, woodwork etc.?	3.78 ± 0.587	3.67 ± 0.584	0.171
Q8: Reading subtitles on TV?	3.85 ± 0.406	3.79 ± 0.435	0.248
Q9: Seeing to engage in an activity/hobby that you are interested in?	3.88 ± 0.427	3.9 ± 0.357	0.724
Q10: Reading small print, such as newspaper articles, items on a menu, telephone directories?	3.78 ± 0.541	3.91 ± 0.318	0.031
Q11: Reading labels/instructions/ingredients/prices such as on: medicine bottles, food packaging?	3.72 ± 0.55	3.83 ± 0.426	0.101
Q12: Reading your post/mail, such as: electric bill, greeting cards, bank statements, letters from friends & family?	3.84 ± 0.48	3.85 ± 0.383	0.878
Q13: Writing and reading your own writing, such as: greeting cards, notes, letters, filling in forms, checks, signing your name?	3.9 ± 0.384	3.88 ± 0.355	0.706
Q14: Seeing the display and keyboard on a computer or calculator?	3.88 ± 0.358	3.91 ± 0.281	0.438
Q15: Seeing the display and keyboard on a mobile or fixed telephone?	3.85 ± 0.406	3.93 ± 0.264	0.151
Q16: Seeing objects close to you and engaging in your hobbies, such as: playing card games, gardening, seeing photographs?	3.93 ± 0.255	3.96 ± 0.248	0.442
Q17: Seeing objects close to you in poor or dim light?	3.59 ± 0.650	3.69 ± 0.644	0.250
Q18: Maintaining focus for prolonged near work?	3.70 ± 0.61	3.78 ± 0.415	0.251
Q19: Conducting near work?	3.83 ± 0.489	3.81 ± 0.393	0.744
Q20: How satisfied are you with your near vision?	4.28 ± 0.912	3.73 ± 0.613	0.011

Grading scale: 1 = yes, very great difficulty, 2 = yes, great difficulty, 3 = yes, some difficulty, 4 = no, no difficulty, 5 = cannot decide (except for question 3 and question 20, for which 1 = very dissatisfied, 2 = generally acceptable, 3 = satisfied, 4 = quite satisfied, 5 = very satisfied).

## Data Availability

The datasets analyzed during the current study are not publicly available but are available from the corresponding author upon reasonable request.

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
