# Peer review of "Evaluation of Life Quality of Patients Submitted to Cataract Surgery with Implantation of Trifocal Intraocular Lenses"

_jpm, 2023, doi:10.3390/jpm13030451_

Round 1
Reviewer 1 Report
In eyes with shorter AL during reading, there may be a greater shift of the IOL toward the cornea. This may account for the better near vision in Brazilian individuals.
[*] What is the main question addressed by the research? Compare the results of operations on two different files.
[*] Is it relevant and interesting? Yes, it is relevant.
[*] How original is the topic? The work is original..
[*] What does it add to the subject area compared with other published
material? The paper compares two different groups in both eye length and use of vision.
[*] Is the paper well written? Yes, it is.
[*] Is the text clear and easy to read? Yes, it is.
[*] Are the conclusions consistent with the evidence and arguments
presented? Yes, they are.
[*] Do they address the main question posed? Yes, it does.
Reviewer 2 Report
Your paper is well written, providing good data and showing the good outcome after this procedure.
It is a well structured and written paper, containing all the necessary steps.
This article presents the quality of life of patients who undergone cataract surgery with Acysof IQ PanOptix Model TFNT00 and the result are satisfactory among the two groups included in the study.
The results and the conclusions contains all the necessary data to understand and use this procedure.
2.In present the indications of using intraarticular HA are for moderate OA which means that actual study does not provide much novelty in the field.
This article sustains the well-known idea that this treatment works better in this stage, this can be considerate a strength of this article but not a novelty. A strength because reiterate and consolidate the idea that HA is efficient in moderate OA.
Author Response
Thank you for your comment! It's a great encouragement to our work!
Reviewer 3 Report
The paper “Evaluation of life quality of patients submitted to cataract surgery with implants of trifocal intraocular lenses” provides a useful addition to the scientific literature on the quality of life and the satisfaction level of patients who have undergone cataract surgery with Acysof IQ PanOptix Model TFNT00 (Alcon Laboratories, Fort 18 Worth, TX) implantation among Brazilian and Chinese patients.
The paper is well written, clear and the discussion supports the presented data. However, I suggest some changes, as follows:
1.Authors: the order of the authors will determine the order of the respective institutions. E.g:
Adilamu Abulimiti5 will have number 3.
2.Chapter 2. Line 76: “.. by surgeon X.C (China) and surgeon Hida W.T. (Braille)."
Please, correct: X. Chen, and.. Brazil?
3.Line 90, 125, 127, 172, 203, 239, 243: to be reviewed
